# Dynamics of Clinical, Radiological, and Biological Expression in Hospitalised COVID-19 Patients and Post-Infectious Mortality: Comparison Between 2024 and 2020

**DOI:** 10.3390/v17050630

**Published:** 2025-04-27

**Authors:** Constantin-Marinel Vlase, Alina Plesea-Condratovici, Anca-Adriana Arbune, Roxana-Elena Goroftei-Bogdan, Cristian Gutu, Manuela Arbune

**Affiliations:** 1Medical Clinic Department, Dunarea de Jos University, 800008 Galati, Romania; constantin.vlase@ugal.ro (C.-M.V.); roxana.goroftei@ugal.ro (R.-E.G.-B.); cristian.gutu@ugal.ro (C.G.); manuela.arbune@ugal.ro (M.A.); 2“Dr. Aristide Serfioti” Military Emergency Hospital, 800150 Galati, Romania; 3Medical Department, Dunarea de Jos University, 800008 Galati, Romania; 4Neurology Department, Fundeni Clinical Institute, 077086 Bucharest, Romania; anca.arbune@icfundeni.ro; 5Multidisciplinary Integrated Center of Dermatological Interface Research (MIC-DIR), “Dunărea de Jos” University, 800008 Galați, Romania; 6Clinic Emergency Children Hospital, 800487 Galati, Romania; 7Infectious Diseases Clinic I, Infectious Diseases Clinic Hospital Galati, 800179 Galati, Romania

**Keywords:** COVID-19, pneumonia, prognosis, clinical variations, 90-day mortality

## Abstract

Clinical manifestations and mortality caused by SARS-CoV-2 change, consistent with circulating viral variants and population immunity. Our retrospective study analyses the hospitalised COVID-19 cases from Romania in 2024 comparative with 2020. The circulation of SARS-CoV-2 in the fourth year of the pandemic led to changes in the characteristics of the hospitalised population, including clinical, biological, and radiological profiles, as well as an increase in 90-day mortality after hospitalisation. The hospitalised population became predominantly older with multiple comorbidities. Notable changes in symptomatology included an increase in the frequency of rhinorrhoea and syncope, while disturbances in taste and smell were less frequent. The mortality rate within the first 90 days increased in 2024 to 9%. Older age and a higher index of chronic comorbidities were associated with a higher frequency of severe forms and deaths within 90 days post-COVID-19. Hospitalisation for COVID-19 may be an unfavourable prognostic factor for the elderly population.

## 1. Introduction

In December 2019, a new virus associated with severe acute respiratory syndrome, named SARS-CoV-2, was identified and rapidly spread globally, leading to a pandemic. It was soon observed that the manifestations of this infection vary from asymptomatic forms to critical forms, with symptoms and severity fluctuating during the illness. The clinical presentation is diverse and can include nasal congestion or runny nose, cough, diarrhoea, fatigue, fever or chills, headache, muscle or body aches, recent loss of taste or smell, nausea or vomiting, difficulty breathing or shortness of breath, and sore throat. The risk of severe progression increases in elderly patients with preexisting conditions, especially obesity, diabetes, and hypertension. Global public health organisations have developed case definitions for COVID-19, included in clinical guidelines and epidemiological surveillance protocols. However, the application of these definitions can be subjective when triaging hospitalised patients [1,2,3].

Although COVID-19 continues to be a public health issue, compared to the first pandemic year, the impact on human health in 2024 has decreased due to post-infectious or post-vaccination immunity in the population, the attenuation of the virulence of circulating strains, and more efficient management of cases. The number of intensive care unit admissions and deaths has decreased compared to the 2020–2022 period, but the information is limited, as these data are not reported by all countries. The emergence of variants with increased virulence and the broad implications of post-COVID syndrome continue to be concerns for healthcare systems [4].

By 2023, the acute phase of the COVID-19 pandemic ended. Moving forward, it is essential to monitor the virus’s ongoing evolution, especially considering the widespread immunity in the global population [5].

Several scenarios remain possible for the evolution of SARS-CoV-2 and human immunity: optimistic, realistic, and pessimistic, as well as a potential ‘reset’ of the pandemic through the emergence of a completely new SARS-CoV-2 virus, originating from an existing animal reservoir or a recently emerged one or through the recombination of two co-infecting variants within the same patient [6].

During the pandemic, authorities in numerous countries reported an ‘excess mortality’ in the population, which decreased in 2023 but remained higher than the pre-pandemic period. COVID-19 can contribute to mortality both directly and indirectly, through its impact on lifestyle and metabolic disorders, accompanied by obesity and diabetes. Extensive studies from clinical trials, as well as sociological and financial studies, have recently drawn attention to concerns within social insurance systems about the persistence of excess mortality for a longer period, projected until 2033, when mortality is expected to decrease by three to four times compared to 2023. Post-COVID effects could impact various causes of morbidity for many years, particularly cardiovascular diseases, which are leading causes of death, requiring careful monitoring [7].

Our study hypothesises that the clinical manifestations, severity, and mortality of hospitalised patients in 2024 have changed compared to the first pandemic year. The objectives of the study are to evaluate the demographic characteristics and clinical and biological manifestations of hospitalised COVID-19 patients in 2024 compared to 2020, as well as the mortality rate in the first 90 days post-COVID-19.

## 2. Materials and Methods

We conducted a retrospective study of COVID-19 hospitalised cases at the Sf. Cuv. Parascheva Clinical Infectious Diseases Hospital in Galați, Romania, during the periods 1 July 2020–31 December 2020 and 1 January 2024–31 December 2024. These time intervals are representative for characterising the morbidity of COVID-19 in the first pandemic year (SARS-CoV-2 Alpha variant) and for observing changes in the clinical–evolutionary expression of infection with derived viral variants over the next four years in hospitalised patients. From the hospital’s database (Hippocrates Medical Management Software), we selected cases of patients aged over 18 years at the time of hospitalisation who were discharged (either sent home or transferred to other medical or sociomedical institutions) and had the diagnostic code U07.1, corresponding to ‘COVID-19, virus identified’ cases [8]. The diagnostic code U07.1 was used when COVID-19 was confirmed by the laboratory criteria, according to the case definition.

From the hospital’s database, we retrospectively collected demographic data (age, sex, residence area, education level, and occupation) and the Charlson comorbidity index score [9,10], as well as the association with other comorbidities such as hypertension and obesity (defined by BMI = weight (kg)/height (m)² ≥ 30 kg/m^2^).

We also collected clinical data such as the duration of symptoms from onset to hospitalisation; the type and frequency of clinical symptoms; vital parameters upon admission; associated radiological phenotypes (interstitial, unilateral, or bilateral opacities); biological markers; the severity of the infection; and reported complications. The evaluation of the biological status included haematological markers (white blood cell count, neutrophil/lymphocyte ratio, haemoglobin, and platelets); D-dimers; C-reactive protein; alanine and aspartate aminotransferases (ALT and AST); and creatine kinase (CK). The management of hospitalised COVID-19 cases was performed according to the current national protocols at the time of patient admission [11,12].

Survival 90 days after discharge was verified using the national health insurance database, recording deceased cases. Statistical analysis was performed using XL-Stat software, version 2020.1. Descriptive statistical analyses were conducted based on the type of variables, calculating the mean, standard deviation, median, extreme values (maximum and minimum), frequency, and distribution. Data comparison was done using Pearson’s tests, 2-tailed Student’s *t*-test (χ^2^), Kruskal–Wallis, Mann–Whitney tests, or correlation and linear regression tests, considering a statistical significance level of *p* < 0.05.

## 3. Results

A total of 721 patients were evaluated in the study, of whom 583 were hospitalised for COVID-19 in 2020 and 138 in 2024.

### 3.1. Demographic Data

The average age of hospitalised COVID-19 patients was 55.66 ± 15.27 years in 2020 compared to 65.13 ± 18.90 years in 2024, with median values of 53 and 70 years, respectively, indicating older patients in 2024 (*p* < 0.001). In both 2020 and 2024, more women were hospitalised, but the difference was not statistically significant compared to men (OR = 1.02; *p* = 0.879). In both periods, patients from urban areas predominated (OR = 4.06; *p* < 0.001).

### 3.2. Medical History and Comorbid Conditions

The age-adjusted Charlson comorbidity index ranged from 0 to 13, with higher values in 2024 compared to 2020, both in terms of the mean (2.01 ± 2.22 vs. 5.07 ± 3.25) and the median (1 vs. 5) (Figure 1).

Except for collagen diseases and AIDS, the other comorbidities included in the Charlson comorbidity index were more prevalent in 2024 (Table 1). Hypertension was reported in 59.6% of patients in 2024 and in 30% of patients in 2020, making it the most common comorbidity among hospitalised COVID-19 patients (Table 1). The prevalence of obesity was also higher in 2020 compared to 2024, although the interpretation of this result is limited due to the lack of anthropometric measurements (weight and waist circumference) in some cases (Table 1).

All patients hospitalised with COVID-19 in 2020 were experiencing their first confirmed episode of the disease and were not vaccinated, as no COVID-19 vaccine was available in Romania at that time. Among the patients hospitalised with COVID-19 in 2024, 2 patients reported a previous episode of COVID-19, and only 12 were vaccinated against COVID-19 (with one to three doses of one of the available vaccine types), but none had received a booster dose in the past year. Thus, only 8.69% of the hospitalised COVID-19 cases had a vaccination history against COVID-19. However, the level of post-infectious natural immunity may be higher due to asymptomatic or oligosymptomatic forms of COVID-19 from previous pandemic waves.

### 3.3. Clinical Manifestations of COVID-19

The duration of symptoms from onset to hospitalisation was significantly longer in the first pandemic year compared to 2024, with an average of 5.87 ± 3.67 days vs. 2.81 ± 2.40 days, and a median of 5 [1;21] vs. 2 [1;18]. COVID-19 symptoms are typically categorised under respiratory viral infections and are characterised by fever; chills; pain syndrome (headache, myalgias, and arthralgias); and respiratory symptoms (dysphagia, dysphonia, rhinorrhoea, cough, and dyspnoea), along with varying digestive (diarrhoea and vomiting) or neurological symptoms. When comparing the clinical picture in 2024 to that of 2020, we observed differences, such as an increase in the frequency of fever, chills, rhinorrhoea, and asthenia, while disturbances in taste and smell, myalgias, and loss of appetite were less frequent (Table 2).

The significance of the changes in symptomatology should be interpreted with caution, considering the higher proportion of elderly patients with neurocognitive disorders, who may express certain symptoms in a distorted manner.

The vital parameters reported during the initial examination at the time of admission for patients showed no significant differences in heart rate, systolic blood pressure, and oxygen saturation. However, diastolic blood pressure and respiratory rate were higher in 2024 compared to the median values of 2020, which were normal (Table A1). The higher respiratory rate in 2020 may be due to the longer duration of the disease from onset to admission compared to 2024, indicating an association with the onset of more common acute pulmonary complications.

### 3.4. Laboratory Findings

In accordance with the updated national protocols, the diagnosis of COVID-19 was based on clinical and epidemiological criteria. RT-PCR-SARS-CoV-2 tests confirmed the laboratory diagnosis in all cases in 2020 but only 16.6% of cases in 2024, when rapid tests, accepted for diagnostic purposes, were also used. Among the known markers with a severe prognostic value for the progression of COVID-19, the neutrophil/lymphocyte ratio (NLR) > 5 and elevated D-dimers levels were more frequently increased in patients hospitalised in 2024, while the CRP levels were higher in 2020, although the difference was not statistically significant (Table A1). The severity and frequency of cytolysis were less pronounced in 2024 compared to 2020, indicating a reduced hepatic impact on the clinical and biological dynamics of COVID-19 (Figure 2).

### 3.5. Radiological Findings

The radiological findings in COVID-19 patients were predominantly characterised by interstitial changes in 2024, in contrast to 2020, when nearly half of the cases had bilateral opacities, corresponding to forms with a potential for severe progression. Complex radiological descriptions, combining interstitial changes with unilateral or bilateral opacities, were noted in both periods, but the proportion of normal radiographs was higher in 2020 compared to 2024 (Table A2).

### 3.6. Clinical Forms, Complications, and Post-COVID-19 Deaths

Pneumonia was the main clinical form of COVID-19, with a higher frequency in 2024, but complications such as respiratory failure and associated hepatitis decreased compared to the first pandemic year (Table A3). The frequency of severe forms and deaths within 90 days of hospitalisation for COVID-19 was higher in 2024 compared to 2020 (Figure A1; Table 3). This evolution can be explained by the more stringent selection of hospitalised cases in 2024, considering the risk of severe progression, with a predominance of older patients and those with a higher comorbid burden. Co-infection with *Clostridioides difficile*, either as an infection that developed in COVID-19 patients or as a preexisting infection upon which COVID-19 was contracted, was significantly more common in 2024 compared to 2020. It served as an additional factor for a severe prognosis in these patients [13].

The average duration of hospitalisation for COVID-19 decreased from 10.77 ± 5.10 days in 2020 to 6.30 ± 3.97 days in 2024 (*p* < 0.001), noting that regulations regarding admission and discharge criteria were modified with the updating of local protocols.

Mortality in the following 90 days after COVID-19 hospitalisation was associated with male sex; a Charlson comorbidity index > 5; elevated CRP (>100 mg/dL); D-dimers (>1000 mg/L); NLR > 5; and the presence of comorbidities such as dementia, chronic heart disease, and co-infection with *Clostridioides difficile*.

### 3.7. Dynamics of COVID-19 Infection and Characteristics of Hospitalised Patients

The circulation of SARS-CoV-2 in the fourth year since the start of the pandemic led to changes in the characteristics of the hospitalised population, including clinical, biological, radiological profiles, and 90-day mortality after hospitalisation. The hospitalised population with COVID-19 became increasingly focused on elderly individuals with multiple comorbidities. The most evident changes in symptomatology included an increase in the frequency of rhinorrhoea and syncope, while disturbances in taste and smell were reported much less frequently. The mortality rate within the first 90 days of COVID-19 hospitalisation increased in 2024. Adverse prognostic factors included male sex; a Charlson comorbidity index greater than 5; elevated levels of prognostic markers (CRP, NLR, and D-dimers); and the presence of dementia or chronic heart disease. Although obesity and diabetes were identified as risk factors for severe COVID-19 progression in 2020, they were not correlated with 90-day survival after COVID-19 hospitalisation (Figure 2).

## 4. Discussion

The dynamics of the epidemiological and clinical criteria observed during the COVID-19 pandemic highlight the evolution of knowledge regarding the new infection at the virological, pathogenic, clinical, and therapeutic levels. This evolution is a result of clinical studies and technological advancements, which have allowed for the improvement of diagnostic, treatment, and prevention methods. The characteristic clinical manifestations of the disease can change depending on factors such as the variation in attack rates across different populations with varying age pyramids, geographic conditions, access to healthcare services, prevalence of comorbidities, and cultural differences that influence the recognition of certain symptoms [14].

The use of updated vaccines and booster doses, as well as the spread of SARS-CoV-2 variants with lower virulence, have contributed to the reduction in morbidity, hospitalisation rates, and COVID-19-related deaths. However, moderate and severe cases continue to be diagnosed, with mortality rates higher than those associated with influenza and other respiratory diseases [15].

Infection with the Omicron variant of SARS-CoV-2 in hospitalised adults was less severe compared to infection with the Delta variant [16,17]. Approximately 20% of hospitalisations during the Omicron period in adults who tested positive for SARS-CoV-2 via RT-PCR were due to non-COVID-19-related conditions, which helps explain the high community transmission of SARS-CoV-2. COVID-19 vaccination was associated with a lower likelihood of hospitalisation in intensive care units (ICUs) during the circulation of the Omicron variant [18,19,20].

Each wave of COVID-19 and its viral variant have led to distinct clinical phenotypes, which have evolved throughout the course of the pandemic. As time passed, the pandemic coronavirus continuously evolved, gradually adapting to environmental conditions and the human immune response. The variation in symptoms presented can be attributed to the influence of new variants and the interference of immunity through vaccination, necessitating a revision of the clinical diagnostic criteria and case definitions, as well as the constant adaptation of healthcare systems to these changes [21].

This reality is also reflected in the results of our study, where the demographic, clinical, biological, and radiological characteristics of COVID-19 changed in hospitalised patients in 2024 compared to the early stages of the pandemic. Older age and a higher index of chronic comorbidities were associated with a higher frequency of severe forms and an increased 90-day mortality rate post-COVID-19, rising from 4% in 2020 to 9% in 2024. The 90-day post-COVID mortality has also been evaluated by other previous retrospective studies. In a UK study at the beginning of the pandemic, 90-day mortality after COVID-19 hospitalisation was 34%, increasing in institutionalised individuals; men; the elderly; and those with chronic comorbidities such as renal, neurological, pulmonary, and cardiac conditions, as well as cumulative comorbidities [22].

In a 2021 study from Malaysia, the 90-day mortality after COVID-19 hospitalisation was 13.5%, correlated with age over 60 years, the need for supplemental oxygen with high flow, and the presence of diabetes [23].

A meta-analysis on morbidity and mortality within the first 90 days after hospitalisation for COVID-19 reported a rehospitalisation rate of 12% based on 295,892 patients and a mortality rate (regardless of cause) of 5% based on 176,920 patients. These data highlight the persistent risks after COVID-19 hospitalisation, including complications and deaths, underscoring the need for the development of monitoring strategies and public health interventions to limit adverse events [24].

### Study Limitations

This study has several limitations, primarily related to the evolution of scientific knowledge and its influence on clinical practice and the management of COVID-19. At the onset of the pandemic, there were no specific treatment protocols in place. By contrast, in 2024, patients received targeted therapies in accordance with an updated national protocol that is aligned with internationally accepted guidelines. Remdesivir continued to be the main antiviral recommended for hospitalised patients, although newer antivirals became available—primarily for outpatient use. Monoclonal antibodies, which were recommended in 2020, were no longer considered effective in 2024. Similarly, anti-interleukin therapies are no longer widely used, as reports of “cytokine storms” have significantly decreased over the past year. Additionally, changes in laboratory diagnostic criteria over time represent a potential source of bias in the study. While all 2020 cases were RT-PCR confirmed, only a subset of 2024 cases underwent RT-PCR testing, with the remainder diagnosed based on rapid antigen testing and clinical/epidemiological criteria in accordance with the updated national guidelines.

Although radiological phenotypes were collected and compared between 2020 and 2024, the analysis did not quantify the extent of radiological involvement. Previous studies have demonstrated that the severity of pulmonary findings on chest imaging is a strong independent predictor of respiratory failure, the need for mechanical ventilation, and mortality in patients with COVID-19 [25]. Quantifying the radiologic severity and analysing its interaction with clinical and biological markers would have provided a more accurate assessment of its prognostic value across the two time periods.

Furthermore, the retrospective design of our study and its restriction to a single clinical centre may limit the generalisability of the findings to other settings or populations with different hospitalisation criteria, healthcare resources, or circulating viral variants.

## 5. Conclusions

The demographic, clinical, biological, and radiological characteristics of COVID-19 are dynamic, with differences observed in hospitalised patients in 2024 compared to the early stages of the pandemic. Older age and a higher index of chronic comorbidities were associated with a higher frequency of severe forms and deaths within 90 days post-COVID-19. Hospitalisation for COVID-19 may be an unfavourable prognostic factor for the elderly population, although this risk appears to be considerably attenuated in individuals with a history of vaccination against COVID-19.

## Figures and Tables

**Figure 1 viruses-17-00630-f001:**
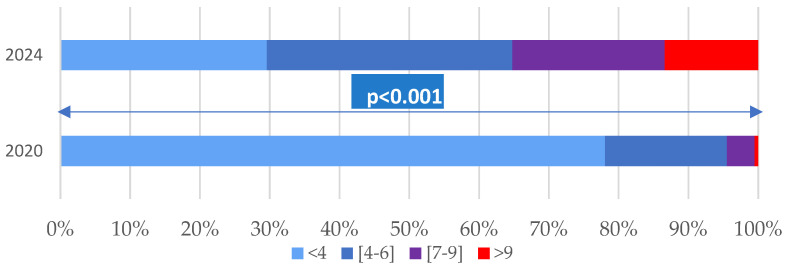
Comparison of the Charlson index in COVID-19 patients hospitalised in 2020 and 2024.

**Figure 2 viruses-17-00630-f002:**
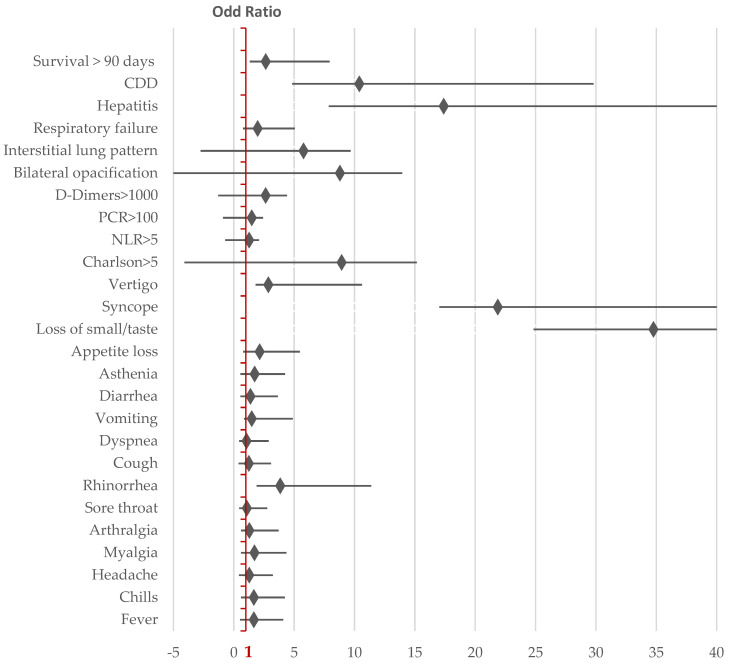
Forest plot illustrating the odds ratios and 95% confidence intervals to identify the differences in COVID-19 characteristics between 2020 and 2024; level of significance *p* < 0.01, two-tailed.

**Table 1 viruses-17-00630-t001:** Comparison of comorbidities in hospitalised COVID-19 patients in 2024 and 2020.

		2024	%	2020	%	OR	CI95	*p* *
Dementia	YES NO	22 115	16.1%	22 561	3.77%	4.878	2.74;8.66	<0.001
Cerebrovascular Accident	YES NO	17 119	12.5%	15 568	2.57%	5.409	2.81;10.409	<0.001
Chronic Heart Faillure	YES NO	74 63	54%	59 524	10.1%	10.43	7.09;15.34	<0.001
Myocardial infarction	YES NO	7 131	5.07%	8 575	1.37%	3.84	1.46;10.06	0.006
Chronic Pulmonary Disease	YES NO	27 110	19.7%	15 568	2.57%	9.29	5.26;16.39	<0.001
Liver disease	YES NO	37 101	26.82%	27 556	4.63%	7.69	4.73;12.52	<0.001
Chronic kidney disease	YES NO	17 121	12.3%	24 559	4.12%	3.27	1.75;6.08	<0.001
Diabetes mellitus	YES NO	25 113	18.1%	60 523	10.3%	1.92	1.16;3.18	0.010
Hemiplegia	YES NO	7 131	5.07%	7 576	1.2%	4.39	1.65;11.70	0.003
Peripheral vascular disease	YES NO	15 123	10.9%	6 577	1.03%	11.72	5.37;25.59	<0.001
Connective tissue disease	YES NO	1 137	0.72%	5 578	0.86%	1.18	0.13;10.20	0.877
Peptic ulcer disease	YES NO	23 115	16.62%	5 578	0.86%	22.11	10.76;45.43	<0.001
AIDS	YES NO	3 135	2.17%	5 578	0.86%	2.56	10.76;45.43	0.184
Tumors/ leukemia lymphoma/	YES NO	19 119	13.8%	27 556	4.63%	3.28	1.82;5.93	<0.001
Hypertenssion	YES NO	81 55	59.6%	175 408	30%	3.43	2.36;4.98	<0.001
Obesity	YES NO	28 70	28.6%	176 269	39.6%	1.63	1.01;2.62	0.042

* Chi-square test.

**Table 2 viruses-17-00630-t002:** Comparison of COVID-19 symptoms in hospitalised patients in 2024 and 2020.

		2024	%	2020	%	OR	CI95	*p* *
Fever	Yes No	86 52	62.3%	290 292	49.8%	1.66	1.14;2.43	0.008
Chills	Yes No	36 102	26.1%	102 481	17.5%	1.66	1.07;2.56	0.021
Headache	Yes No	41 97	29.7%	144 439	24.7%	1.28	0.85;1.94	0.225
Myalgia	Yes No	31 106	22.6%	195 388	33.4%	1.71	1.11;2.64	0.014
Arthralgia	Yes No	14 123	10.2%	47 536	8.06%	1.29	0.69;2.42	0.507
Sore throat	Yes No	31 107	22.5%	123 460	21.1%	1.08	0.639;1.693	0.724
Rhinorrhea	Yes No	15 118	10.9%	18 565	3.9%	3.85	1.97;7.53	<0.001
Cough	Yes No	79 59	57.2%	217 366	62.8%	1.25	0.86;1.835	0.229
Dyspnea	Yes No	19 119	13.8%	76 507	13%	1.06	0.62;1.82	0.819
Vomiting	Yes No	8 130	5.8%	23 560	3.95%	1.49	0.65;3.40	0.334
Diarrhea	Yes No	25 112	18.2%	81 502	13.9%	1.38	0.84;2.26	0.195
Asthenia	Yes No	72 66	52.2%	225 358	38.6%	1.73	1.19;2.51	0.003
Appetite loss	Yes No	28 108	20.4%	207 376	35.5%	2.14	1.37;3.33	0.007
Loss of small/taste	Yes No	1 137	0.72%	118 465	20.2%	34.76	9.93;121.64	<0.001
Syncope	Yes No	5 133	3.62%	1 582	0.17%	21.87	4.84;98.72	<0.001
Vertigo	Yes No	4 134	2.9%	46 537	7.89%	2.86	1.06;7.76	0.037

* Chi-square test.

**Table 3 viruses-17-00630-t003:** Ninety-day mortality after COVID-19 hospitalisation (all patients).

90 Days After-COVID-19 Hospitalisation		Survive N1 = 687	Death N2 = 35	OR	CI95	*p*
Male	Yes No	396 326	23 12	2.42	1.21;4.86	0.012
Charlson>5	Yes No	553 134	22 13	6.98	3.73;13.06	<0.001
*Clostridioides difficile * Diarrhoea	Yes No	26 661	9 26	8.80	4.26;18.14	<0.001
CRP > 100 mg/dL	Yes No	95 592	18 17	6.59	3.55;12.25	<0.001
NLR > 5	Yes No	129 558	19 16	5.13	2.73;9.66	<0.001
D-Dimers > 1000 mg/L	Yes No	131 556	22 13	4.28	3.84;13.41	<0.001
Dementia	Yes No	37 650	7 28	4.39	1,92;9.99	<0.001
Chronic Heart Disease	Yes No	120 566	13 22	2.78	1.40;5.54	0.003
Diabetes	Yes No	80 607	6 29	1.56	0.63;3.87	0.327
Obesity	Yes No	198 329	6 11	0.906	0.33;2.48	0.848

## Data Availability

This study was conducted in accordance with the Declaration of Helsinki, for studies involving humans and approved by the Institutional Board of Clinic Hospital for Infectious Diseases Galati (No.1/6; date of 21 January 2025).

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
