# Peer review of "Dynamics of Clinical, Radiological, and Biological Expression in Hospitalised COVID-19 Patients and Post-Infectious Mortality: Comparison Between 2024 and 2020"

_viruses, 2025, doi:10.3390/v17050630_

Round 1
Reviewer 1 Report
Comments and Suggestions for Authors
A cordial greeting, I congratulate the authors for their paper, the research question is important, however, they must recognize the biases of the research in the discussion, it is important to clarify if there were significant changes in the treatment during these two moments evaluated, I consider that since the cases of COVID 19 are not corroborated by the laboratory in most of the events, this generates an important bias in the analysis, on the other hand, knowing in more detail the vaccination status for COVID 19 can be very important for the conclusions. The literature has shown different findings between regions regarding the manifestation and outcomes of COVID 19 infection after the first year of the pandemic, and its explanation has been multiple and contradictory, this new evidence adds to the available literature and this will be very useful for subsequent analysis.
Author Response
23.04.2025
Answer to Reviewer 1
Comments and Suggestions for Authors
A cordial greeting, I congratulate the authors for their paper, the research question is important, however, they must recognize the biases of the research in the discussion, it is important to clarify if there were significant changes in the treatment during these two moments evaluated, I consider that since the cases of COVID 19 are not corroborated by the laboratory in most of the events, this generates an important bias in the analysis, on the other hand, knowing in more detail the vaccination status for COVID 19 can be very important for the conclusions. The literature has shown different findings between regions regarding the manifestation and outcomes of COVID 19 infection after the first year of the pandemic, and its explanation has been multiple and contradictory, this new evidence adds to the available literature, and this will be very useful for subsequent analysis.
A: We sincerely thank you for the positive feedback and the valuable comments.
We have revised the manuscript by adding a Study Limitations section, where we discuss the potential biases of the research.
Several limitations are primarily related to the evolution of scientific knowledge and its influence on clinical practice and the management of COVID-19. We acknowledged that the early phase of the pandemic was characterized by the absence of specific treatment protocols for COVID-19, whereas in 2024, patients benefited from targeted therapies based on an updated national protocol (see new reference no. 26), aligned with internationally recognized guidelines. Remdesivir continued to be the main antiviral recommended for hospitalized patients, although newer antivirals became available—primarily for outpatient use. Monoclonal antibodies, which were recommended in 2020, are no longer considered effective in 2024. Similarly, anti-interleukin therapies are no longer widely used, as reports of "cytokine storms" have significantly decreased over the past year.
We fully agree that laboratory confirmation of COVID-19 cases is essential to ensure diagnostic accuracy and to minimize classification bias. In this context, we would like to clarify that all cases in the 2020 cohort were confirmed by RT-PCR, in accordance with the national diagnostic protocol in place at the time. For the 2024 cohort, due to updates in diagnostic criteria and national clinical practice, a proportion of cases were confirmed using rapid antigen tests, which are accepted as valid diagnostic tools under the current National Protocol for COVID-19 Treatment (Order no. 1274/2024 – see reference 26). Although only 16.6% of the 2024 cases were confirmed by RT-PCR, all patients met the clinical and epidemiological criteria for COVID-19 diagnosis, as defined by both national and international guidelines. This clarification has now been included in the manuscript, and we have also acknowledged the potential for diagnostic variability as a limitation in the revised Discussion section.
Furthermore, we have updated the Conclusions section to reflect the possible influence of prior COVID-19 vaccination on clinical outcomes. Specifically, we now emphasize that although hospitalization remains a negative prognostic factor in elderly patients, this risk appears to be significantly reduced among those with a history of COVID-19 vaccination.
We hope that the revised version of our manuscript meets your requirements and will be considered for publication.
Kind regards,
Manuela Arbune
Reviewer 2 Report
Comments and Suggestions for Authors
I reviewed the viruses-3569481 manuscript. The authors conducted a retrospective comparison of the characteristics of the hospitalized patients with COVID-19 between 2020 and 2024. They reached at a number of significant findings regarding the evolution of COVID-19 pandemic and apart from some drawbacks, it could be suitable -in my opinion- for scientific publication. The most important ones are the following:
1) The authors should include in the 'Discussion" section a paragraph addressing the study limitations. In this paragraph they should discuss:
-the fact that COVID-19 diagnosis was relied on the U07.1 code, rather than a PCR swab test.
-the fact that the authors did not evaluate the extent of the radiological findings, a characteristic that it has been associated with the development of respiratory failure, the need for mechanical ventilation and overall mortality in COVID-19 patients. In this regard the authors should include the following reference: "Pappas AG, Panagopoulos A, Rodopoulou A, Alexandrou M, Chaliasou AL, Skianis K, Kranidioti E, Chaini E, Papanikolaou I, Kalomenidis I. Moderate COVID-19: Clinical Trajectories and Predictors of Progression and Outcomes. J Pers Med. 2022 Sep 8;12(9):1472. doi: 10.3390/jpm12091472."
2) The 2 paragraphs from line 244 to line 253 ("Recent studies have shown ..... in differentiating the two viral infection") are not relevant with the study and in my opinion should be removed.
Author Response
23.04.2025
Answer to Reviewer 2
I reviewed the viruses-3569481 manuscript. The authors conducted a retrospective comparison of the characteristics of the hospitalized patients with COVID-19 between 2020 and 2024. They reached at a number of significant findings regarding the evolution of COVID-19 pandemic and apart from some drawbacks, it could be suitable -in my opinion- for scientific publication. The most important ones are the following:
1) The authors should include in the 'Discussion" section a paragraph addressing the study limitations. In this paragraph they should discuss:
-the fact that COVID-19 diagnosis was relied on the U07.1 code, rather than a PCR swab test.
-the fact that the authors did not evaluate the extent of the radiological findings, a characteristic that it has been associated with the development of respiratory failure, the need for mechanical ventilation and overall mortality in COVID-19 patients. In this regard the authors should include the following reference: "Pappas AG, Panagopoulos A, Rodopoulou A, Alexandrou M, Chaliasou AL, Skianis K, Kranidioti E, Chaini E, Papanikolaou I, Kalomenidis I. Moderate COVID-19: Clinical Trajectories and Predictors of Progression and Outcomes. J Pers Med. 2022 Sep 8;12(9):1472. doi: 10.3390/jpm12091472."
A1. Thank you for this insightful comment. We have added a specific statement regarding the use of code U07.1, according to the International Classification of Diseases 11th Revision (ICD-11).
We have included in the 'Discussion" section a paragraph addressing the study limitations, including the laboratory diagnostic criteria and radiological findings, adding the new reference.
2) The 2 paragraphs from line 244 to line 253 ("Recent studies have shown ..... in differentiating the two viral infection") are not relevant with the study and in my opinion should be removed.
A2. We removed the 2 paragraphs from line 244 to line 253.
Thank you very much for your valuable comments and suggestions.
We hope that the revised manuscript to be improved and to be accepted for publication.
Kind regards,
Manuela Arbune